# Fundamental limits to learning closed-form mathematical models from data

Oscar Fajardo-Fontiveros [1,5], Ignasi Reichardt [1,2,5], Harry R. De Los Ríos [3], Jordi Duch [3], Marta Sales-Pardo [1] ✉ & Roger Guimerà [1,4] ✉

Given a finite and noisy dataset generated with a closed-form mathematical model, when is it possible to learn the true generating model from the data alone? This is the question we investigate here. We show that this model-learning problem displays a transition from a low-noise phase in which the true model can be learned, to a phase in which the observation noise is too high for the true model to be learned by any method. Both in the low-noise phase and in the high-noise phase, probabilistic model selection leads to optimal generalization to unseen data. This is in contrast to standard machine learning approaches, including artificial neural networks, which in this particular problem are limited, in the low-noise phase, by their ability to interpolate. In the transition region between the learnable and unlearnable phases, generalization is hard for all approaches including probabilistic model selection.

For a few centuries, scientists have described natural phenomena by means of relatively simple mathematical models such as Newton's law of gravitation or Snell's law of refraction. Sometimes, they arrived to these models deductively, starting from fundamental considerations; more frequently, however, they derived the models inductively from data. With increasing amounts of data available for all sorts of (natural and social) systems, one may argue that we are now in a position to inductively uncover new interpretable models for these systems. To this end, machine learning approaches that can automatically uncover closed-form models from data have been recently developed[1–6]. (Here and throughout this article, we refer to closed-form models as those that are expressed in terms of a relatively small number of basic functions, such as addition, multiplication, trigonometric functions, etc.) In physics alone, such approaches have been applied successfully to quantum systems[7], non-linear and chaotic systems[2,4], fluid mechanics[8], and astrophysics[9], among others[6].

A central assumption implicit in these approaches is that, given data, it is always possible to identify the correct underlying model. Here, we investigate the validity of this assumption. In particular, consider a dataset $D = \{(y_i, \mathbf{x}_i)\}$, with $i = 1, ..., N$, generated using the closed-form model $m^*$, so that $y_i = m^*(\mathbf{x}_i, \theta^*) + \epsilon_i$ with $\theta^*$ being the

parameters of the model, and $\epsilon_i$ a random unbiased observation noise drawn from the normal distribution with variance $s_\epsilon^2$. The assumption of Gaussian noise is standard in regression and symbolic regression problems. In principle, one could assume other noise structures (for example, multiplicative noise) or even more general likelihoods, but these would be hard to justify in the context of regression and symbolic regression/model discovery. The question we are interested in is: Assuming that $m^*$ can be expressed in closed form, when is it possible to identify it as the true generating model among all possible closed-form mathematical models, for someone who does not know the true model beforehand? Note that our focus is on learning the structure of the model $m^*$ and not the values of the parameters $\theta^*$, a problem that has received much more attention from the theoretical point of view[10]. Additionally, we are interested in situations in which the dimension of the feature space $\mathbf{x} \in \mathbb{R}^k$ is relatively small (compared to typical feature spaces in machine learning settings), which is the relevant regime for symbolic regression and model discovery.

To address the model-learning question above, we formulate the problem of identifying the true generating model probabilistically, and show that probabilistic model selection is quasi-optimal at generalization, that is, at making predictions about unobserved data.

[1]Department of Chemical Engineering, Universitat Rovira i Virgili, Tarragona 43007, Catalonia. [2]Department of Mechanical Engineering, Universitat Rovira i Virgili, Tarragona 43007, Catalonia. [3]Department of Computer Science and Mathematics, Universitat Rovira i Virgili, Tarragona 43007, Catalonia. [4]ICREA, Barcelona 08010, Catalonia. [5]These authors contributed equally: Oscar Fajardo-Fontiveros, Ignasi Reichardt. ✉e-mail: marta.sales@urv.cat; roger.guimera@urv.cat

This is in contrast to standard machine learning approaches, which, in this case, are suboptimal in the region of low observation noise. We then investigate the transition occurring between: (i) a learnable phase at low observation noise, in which the true model can in principle be learned from the data; and (ii) an unlearnable phase, in which the observation noise is too large for the true model to be learned from the data by any method. Finally, we provide an upper bound for the noise at which the learnability transition takes place, that is, the noise beyond which the true generating model cannot be learned by any method. This bound corresponds to the noise at which most plausible a priori models, which we call trivial, become more parsimonious descriptions of the data than the true generating model. Despite the simplicity of the approach, the bound provides a good approximation to the actual transition point.

## Results
### Probabilistic formulation of the problem
The complete probabilistic solution to the problem of identifying a model from observations is encapsulated in the posterior distribution over models $p(m|D)$. The posterior gives the probability that each model $m = m(\mathbf{x}, \theta)$, with parameters $\theta$, is the true generating model given the data $D$. Again, notice that we are interested on the posterior over model structures $m$ rather than model parameters $\theta$; thus, we obtain $p(m|D)$ by marginalizing $p(m, \theta|D)$ over possible parameter values $\Theta$

$$p(m|D) = \int_\Theta d\theta\, p(m,\theta|D)$$
$$= \frac{1}{p(D)} \int_\Theta d\theta\, p(D|m,\theta)\, p(\theta|m)\, p(m), \quad (1)$$

where $p(D|m, \theta)$ is the model likelihood, and $p(\theta|m)$ and $p(m)$ are the prior distributions over the parameters of a given model and the models themselves, respectively. The posterior over models can always be rewritten as

$$p(m|D) = \frac{\exp[-\mathcal{H}(m)]}{Z}, \quad (2)$$

with $Z = p(D) = \sum_{\{m\}} \exp[-\mathcal{H}(m)]$ and

$$\mathcal{H}(m) = -\ln p(D,m)$$
$$= -\ln\left[\int_\Theta d\theta\, p(D|m,\theta)\, p(\theta|m)\right] - \ln p(m). \quad (3)$$

Although the integral over parameters in Eq. (3) cannot, in general, be calculated exactly, it can be approximated as[5,11]

$$\mathcal{H}(m) \approx \frac{B(m)}{2} - \ln p(m), \quad (4)$$

where $B(m)$ is the Bayesian information criterion (BIC) of the model[11]. This approximation results from using Laplace's method to integrate the distribution $p(D|m, \theta)p(\theta|m)$ over the parameters $\theta$. Thus, the calculation assumes that: (i) the likelihood $p(D|m, \theta)$ is peaked around $\theta^* = \arg\max_\theta p(D|m,\theta)$, so that it can be approximated by a Gaussian around $\theta^*$; (ii) the prior $p(\theta|m)$ is smooth around $\theta^*$ so that it can be assumed to be approximately constant within the Gaussian. Unlike other contexts, in regression-like problems these assumptions are typically mild.

Within an information-theoretic interpretation of model selection, $\mathcal{H}(m)$ is the description length, that is, the number of nats (or bits if one uses base 2 logarithms instead of natural logarithms) necessary to jointly encode the data and the model with an optimal encoding[12]. Therefore, the most plausible model—that is, the model with maximum $p(m|D)$—has the minimum description length, that is, compresses the data optimally.

The posterior in Eq. (2) can also be interpreted as in the canonical ensemble in statistical mechanics, with $\mathcal{H}(m)$ playing the role of the energy of a physical system, models playing the role of configurations of the system, and the most plausible model corresponding to the ground state. Here, we sample the posterior distribution over models $p(m|D)$ by generating a Markov chain with the Metropolis algorithm, as one would do for physical systems. To do this, we use the "Bayesian machine scientist" introduced in ref. [5] (Methods). We then select, among the sampled models, the most plausible one (that is, the maximum a posteriori model, the minimum description length model, or the ground state, in each interpretation).

### Probabilistic model selection yields quasi-optimal predictions for unobserved data
Probabilistic model selection as described above follows directly (and exactly, except for the explicit approximation in Eq. (4)) from the postulates of Cox's theorem[13,14] and is therefore (Bayes) optimal when models are truly drawn from the prior $p(m)$. In particular, the minimum description length (MDL) model is the most compressive and the most plausible one, and any approach selecting, from $D$, a model that is not the MDL model violates those basic postulates.

We start by investigating whether this optimality in model selection also leads to the ability of the MDL model to generalize well, that is, to make accurate predictions about unobserved data. We show that the MDL model yields quasi-optimal generalization despite the fact that the best possible generalization is achieved by averaging over models[5], and despite the BIC approximation in the calculation of the description length (Fig. 1). Specifically, we sample a model $m^*$ from the prior $p(m)$ described in ref. [5] and in the Methods; for a model not drawn from the prior, see Supplementary Text and Fig. S1. From this model, we generate synthetic datasets $D = \{(y_i, \mathbf{x}_i)\}$, $i = 1, \ldots, N$ (where $y_i = m^*(\mathbf{x}_i, \theta^*) + \epsilon_i$ and $\epsilon_i \sim \text{Gaussian}(0, s_\epsilon)$) with different number of points $N$ and different levels of noise $s_\epsilon$. Then, for each dataset $D$, we sample models from $p(m|D)$ using the Bayesian machine scientist[5], select the MDL model among those sampled, and use it to make predictions on a test dataset $D'$.

Because $D'$ is, like $D$, subject to observation noise, the irreducible error is $s_\epsilon$, that is, the root mean squared error (RMSE) of predictions cannot be, on average, smaller than $s_\epsilon$. As we show in Fig. 1, the predictions of the MDL model achieve this optimal prediction limit except for small $N$ and some intermediate values of the observation noise. This is in contrast to standard machine learning algorithms, such as artificial neural networks. These algorithms achieve the optimal prediction error at large values of the noise, but below certain values of $s_\epsilon$ the prediction error stops decreasing and predictions become distinctly suboptimal (Fig. 1).

In the limit $s_\epsilon \to \infty$ of high noise, all models make predictions whose errors are small compared to $s_\epsilon$. Thus, the prediction error is similar to $s_\epsilon$ regardless of the chosen model, which also means that it is impossible to correctly identify the model that generated the data. Conversely, in the $s_\epsilon \to 0$ limit, the limiting factor for standard machine learning methods is their ability to interpolate between points in $D$, and thus the prediction error becomes independent of the observation error. By contrast, because Eqs. (2)–(4) provide consistent model selection, in this limit the MDL should coincide with the true generating model $m^*$ and interpolate perfectly. This is exactly what we observe—the only error in the predictions of the MDL model is, again, the irreducible error $s_\epsilon$. Therefore, our observations show that probabilistic model selection leads to quasi-optimal generalization in the limits of high and low observation noise.

### Phenomenology of the learnability transition
Next, we establish the existence of the learnable and unlearnable regimes, and clarify how the transition between them happens. Again,

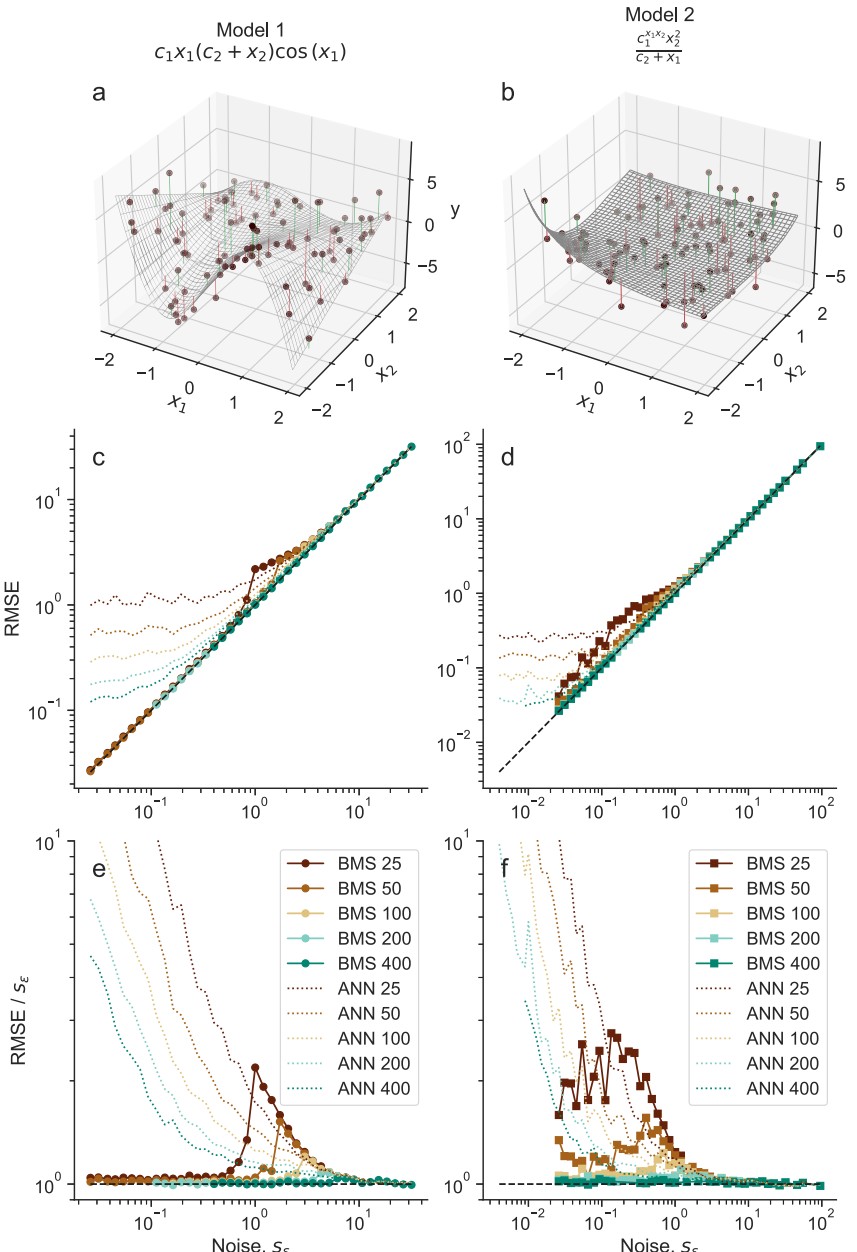

**Fig. 1 | Probabilistic model selection makes quasi-optimal predictions about unobserved data.** We select two models $m^*$, whose expressions are shown at the top of each column. **a**, **b** From each model, we generate synthetic datasets $D$ with $N$ points (shown, $N = 100$) and different levels of noise $s_\epsilon$ (shown, $s_\epsilon = 1$). Here and throughout the article, the values of the independent variables $x_1$ and $x_2$ are generated uniformly at random in $[-2, 2]$. Vertical lines show the observation error $\epsilon_i$ for each point in $D$. For a model not drawn from the prior and data generated differently, see Supplementary Fig. S1. **c**, **d** For each dataset $D$ (with dataset sizes $N \in \{25, 50, 100, 200, 400\}$), we sample models from $p(m|D)$ using the Bayesian

machine scientist[5], select the MDL model (maximum $p(m|D)$) among those sampled, and use this model to make predictions on a test dataset $D'$, generated exactly as $D$. We show the prediction root mean squared error (RMSE) of the MDL model on $D'$ as a function of $N$ and $s_\epsilon$. For comparison, we also show the predictions from an artificial neural network (ANN, dotted lines; Methods). Since $s_\epsilon$ is the irreducible error, predictions on the diagonal RMSE = $s_\epsilon$ are optimal. **e**, **f** We plot the prediction RMSE scaled by the irreducible error $s_\epsilon$; optimal predictions satisfy RMSE/$s_\epsilon = 1$ (dashed line).

we generate synthetic data $D$ using a known model $m^*$ as in the previous section, and sample models from the posterior $p(m|D)$. To investigate whether a model is learnable we consider the ensemble of sampled models (Fig. 2); if no model in the ensemble is more plausible than the true generating model (that is, if the true model is also the MDL model), then we conclude that the model is learnable. Otherwise, the model is unlearnable because, lacking any additional information about the generating model, it is impossible to identify it; other models provide more parsimonious descriptions of the data $D$.

To this end, we first consider the gap $\Delta\mathcal{H}(m) = \mathcal{H}(m) - \mathcal{H}(m^*)$ between the description length of each sampled model $m$ and that of the true generating model $m^*$ (Fig. 2a). For low observation error $s_\epsilon$, all sampled models have positive gaps, that is, description lengths that are longer than or equal to the true model. This indicates that the most plausible model is the true model; therefore, the true model is learnable. By contrast, when observation error grows ($s_\epsilon \gtrsim 0.6$ in Fig. 2a), some of the models sampled for some training datasets $D$ start to have negative gaps, which means that, for those datasets, these models

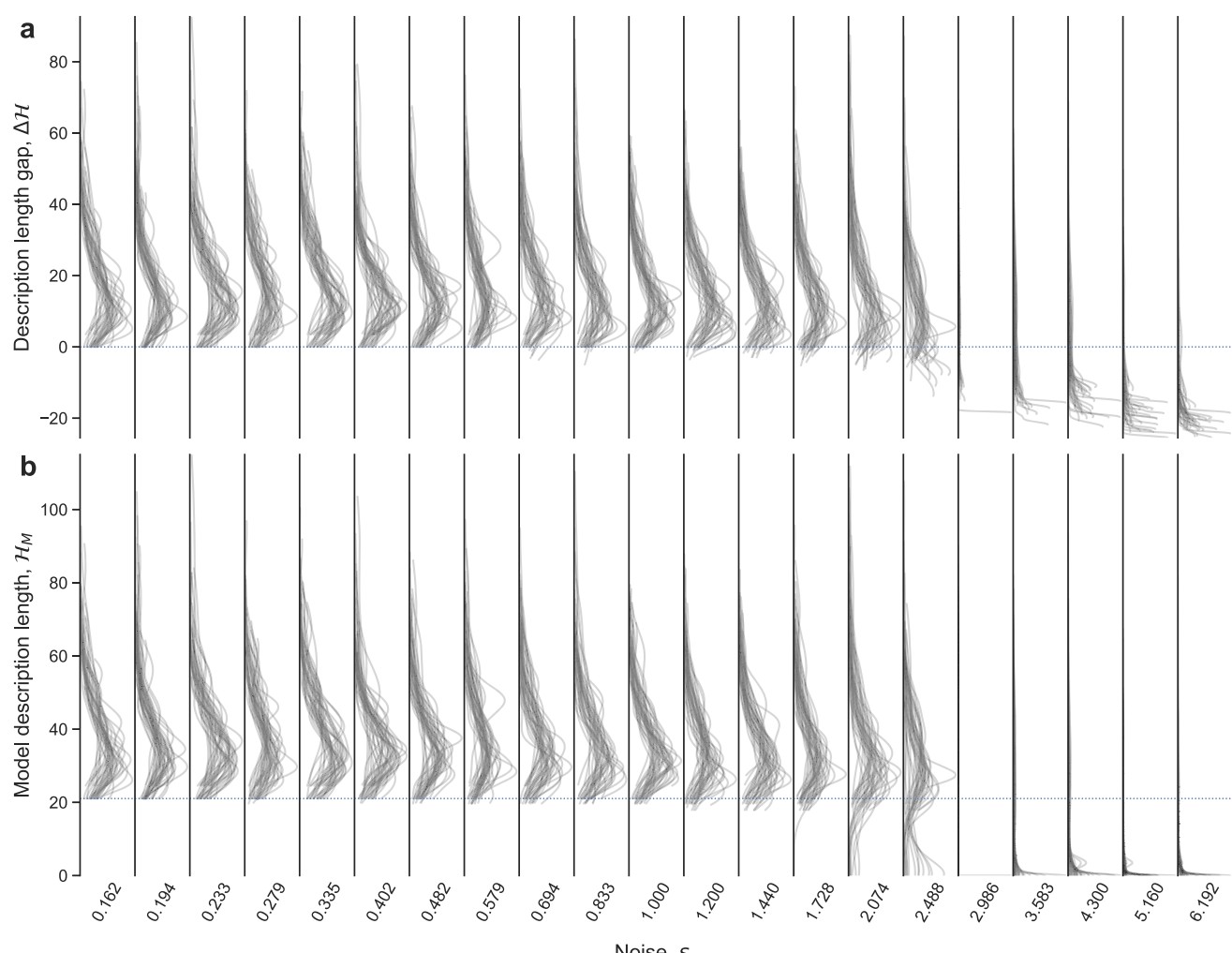

**Fig. 2 | Phenomenology of the learnability transition.** Using "Model 1" in Fig. 1, we generate 40 different datasets $D$ (with $N = 100$) for each observation noise $s_\epsilon$. As before, the values of the independent variables $x_1$ and $x_2$ are generated uniformly at random in $[-2, 2]$. For each dataset, we sample models from the posterior $p(m|D)$ and obtain the distribution of certain properties of the sampled models. **a** Distribution of description length gaps $\Delta\mathcal{H}(m) = \mathcal{H}(m) - \mathcal{H}(m^*)$; each line corresponds to a different dataset $D$. The true generating model has $\Delta\mathcal{H}(m) = 0$, indicated by a horizontal dashed line. Positive gaps correspond to models $m$ that are less plausible than the true generating model $m^*$, and vice versa. When there is a model $m$ with negative gap for a given dataset $D$, the true model is unlearnable from that dataset because $m^*$ is not the most plausible model given the data. **b** Distribution of model description lengths $\mathcal{H}_M(m) = -\ln p(m)$ for each dataset $D$. The $\mathcal{H}_M(m^*)$ of the true generating model is indicated by a horizontal dashed line. Without lost of generality, the $\mathcal{H}_M(m^c)$ of the most plausible model a priori $m^c = \arg\max_m p(m)$, or trivial model, is set to $\mathcal{H}_M(m^c) = 0$.

become more plausible than the true model. When this happens, the true model becomes unlearnable. For even larger values of $s_\epsilon$ ($s_\epsilon > 2$), the true model is unlearnable for virtually all training sets $D$.

To better characterize the sampled models, we next compute description length of the model; that is, the term $\mathcal{H}_M(m) = -\ln p(m)$ in the description length, which measures the complexity of the model regardless of the data (Fig. 2b). For convenience we set an arbitrary origin for $\mathcal{H}_M(m)$ so that $\mathcal{H}_M(m^c) = 0$ for the models $m^c = \arg\max_m p(m) = \arg\min_m \mathcal{H}_M(m)$ that are most plausible a priori, which we refer to as *trivial* models; all other models have $\mathcal{H}_M(m) > 0$. We observe that, for low observation error, most of the sampled models are more complex than the true model. Above a certain level of noise ($s_\epsilon \gtrsim 3$), however, most sampled models are trivial or very simple models with $\mathcal{H}_M(m) \approx 0$. These trivial models appear suddenly—at $s_\epsilon < 2$ almost no trivial models are sampled at all.

Altogether, our analysis shows that for low enough observation error the true model is always recovered. Conversely, in the opposite limit only trivial models (those that are most plausible in the prior distribution) are considered reasonable descriptions of the data.

Importantly, our analysis shows that trivial models appear suddenly as one increases the level of noise, which suggests that the transition to the unlearnable regime may be akin to a phase transition driven by changes in the model plausibility (or description length) landscape, similarly to what happens in other problems in inference, constraint satisfaction, and optimization[10,15].

To further probe whether that is the case, we analyze the transition in more detail. Our previous analysis shows that learnability is a property of the training dataset $D$, so that, for the same level of observation noise, some instances of $D$ enable learning of the true model whereas others do not. Thus, by analogy with satisfiability transitions[15], we define the learnability $\rho(s_\epsilon)$ as the fraction of training datasets $D$ at a given noise $s_\epsilon$ for which the true generating model is learnable. In Fig. 3, we show the behavior of the learnability $\rho(s_\epsilon)$ for different sizes $N$, for the two models in Fig. 1 (see Supplementary Fig. S1 for another model). Consistent with the qualitative description above, we observe that the learnability transition occurs abruptly at a certain value of the observation noise; the transition shifts towards higher values of $s_\epsilon$ with increasing size of $D$.

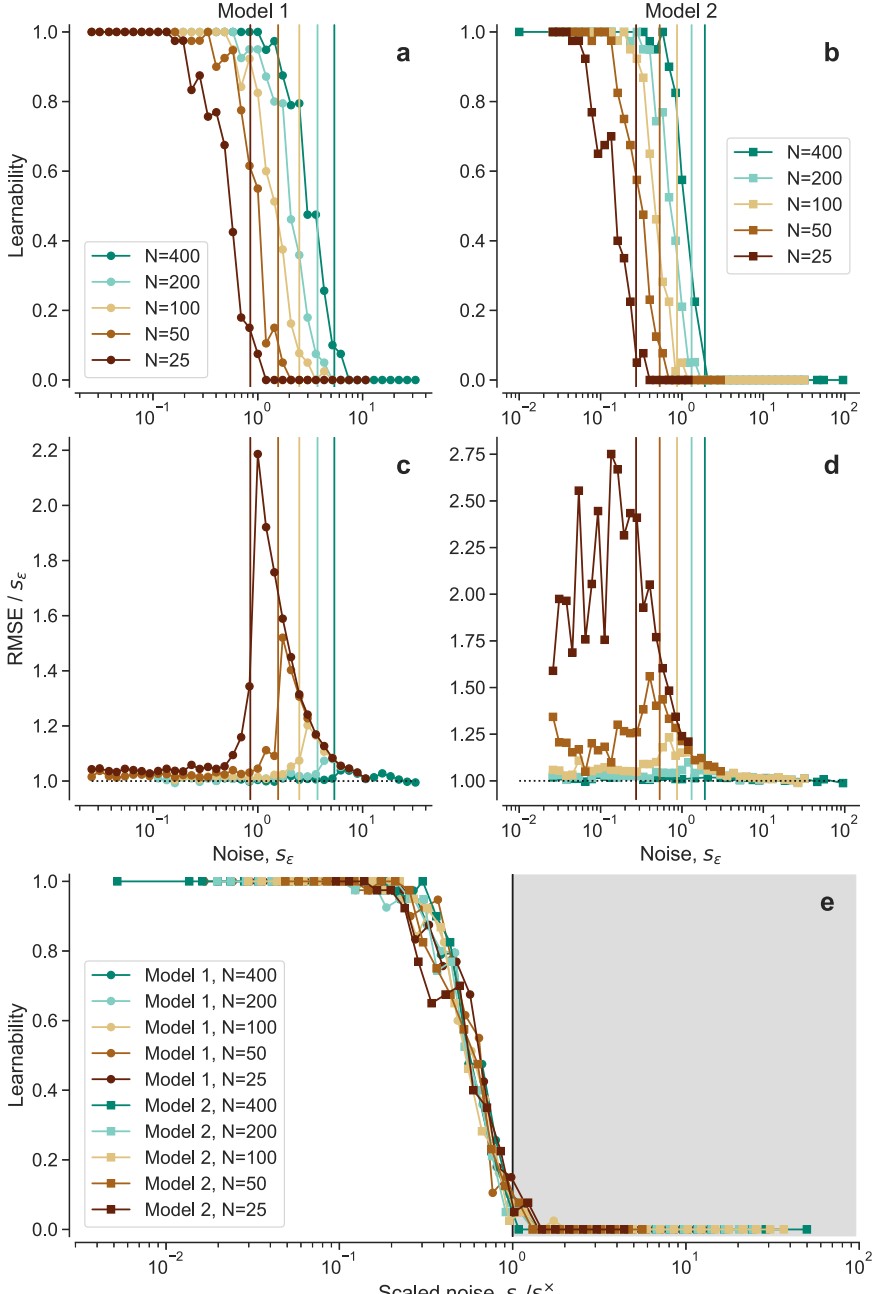

**Fig. 3 | Learnability transition and scaling. a, b** For each of the two models in Fig. 1, we represent the learnability $\rho(s_\varepsilon)$, that is, the fraction of datasets $D$ for which the true model $m^*$ is the most plausible one, and thus can be learned from the data. Vertical lines are estimates of the learnability transition point $s_\varepsilon^\times$ from Eq. (8). **c, d** As in Fig. 1e, f, we represent the scaled prediction root mean squared error (RMSE) for the MDL model. The peak in the scaled RMSE coincides with the learnability transition point. **e** Learnability as a function of the scaled noise $s_\varepsilon/s_\varepsilon^\times$ for all learnability curves (all values of $N$ and both models). The gray region $s_\varepsilon/s_\varepsilon^\times > 1$ identifies the unlearnable phase.

Remarkably, the difficult region identified in the previous section, in which prediction error deviates from the theoretical irreducible error $s_\varepsilon$, coincides with the transition region. Therefore, our results paint a picture with three qualitatively distinct regimes: a learnable regime in which the true model can always be learned and predictions about unobserved data are optimal; an unlearnable regime in which the true model can never be learned but predictions are still optimal because error is driven by observation noise and not by the model itself; and a transition regime in which the true model is learnable only sometimes, and in which predictions are, on average, suboptimal. This, again, is reminiscent of the hard phase in satisfiability transitions and of other statistical learning problems[15–17].

**Learnability conditions**

Finally, we obtain an upper bound to the learnability transition point by assuming that all the phenomenology of the transition can be explained by the competition between just two minima of the description length landscape $\mathcal{H}(m)$: the minimum corresponding to the true model $m^*$; and the minimum corresponding to the trivial model $m^c$ that is most plausible a priori

$$m^c = \arg\max_m p(m) = \arg\min_m \mathcal{H}_M(m). \tag{5}$$

As noted earlier, $m^c$ is such that $\mathcal{H}_M(m^c) = 0$ by our choice of origin, and for our choice of priors it corresponds to trivial models without any

operation, such as $m(\mathbf{x}) = $ const. or $m(\mathbf{x}) = x_1$. (Note that, although the origin of descriptions lengths is not arbitrary, this choice is innocuous as long as we are only concerned with comparisons between models.) Below, we focus on one of these models $m(\mathbf{x}) = $ const., but our arguments (not the precise calculations) remain valid for any other choice of prior and the corresponding $m^c$.

As we have also noted above, the true model is learnable when the description length gap $\Delta\mathcal{H}(m) = \mathcal{H}(m) - \mathcal{H}(m^*)$ is strictly positive for all models $m \neq m^*$. Conversely, the true model $m^*$ becomes unlearnable when $\Delta\mathcal{H}(m) = 0$ for some model $m \neq m^*$. As per our assumption that the only relevant minima are those corresponding to $m^*$ and $m^c$, we therefore postulate that the transition occurs when the description length gap of the trivial model becomes, on average (over datasets), $\Delta\mathcal{H}(m^c) = 0$. In fact, when $\mathcal{H}(m^c) \leq \mathcal{H}(m^*)$ the true model is certainly unlearnable; but other models $m$ could fulfill the condition $\mathcal{H}(m) \leq \mathcal{H}(m^*)$ earlier, thus making the true model unlearnable at lower observation noises. Therefore, the condition $\Delta\mathcal{H}(m^c) = 0$ yields an upper bound to the value of the noise at which the learnability transition happens. We come back to this issue later, but for now we ignore all potential intermediate models.

Within the BIC approximation, the description length of each of these two models is ("Methods" section)

$$\mathcal{H}(m^*) = \frac{N}{2}\left[\log 2\pi \langle \epsilon^2 \rangle_D + 1\right] + \frac{k^* + 1}{2}\log N - \log p(m^*) \qquad (6)$$

$$\mathcal{H}(m^c) = \frac{N}{2}\left[\log 2\pi \left(\langle \epsilon^2 \rangle_D + \langle \delta^2 \rangle_D\right) + 1\right] + \log N - \log p(m^c), \qquad (7)$$

where $k^*$ is the number of parameters of the true model, $\delta_i = m_i^c - m_i^*$ is the reducible error of the trivial model, and the averages $\langle \cdots \rangle_D$ are over the observations $i$ in $D$. Then, over many realizations of $D$, the transition occurs on average at

$$s_\epsilon^\times = \left[\frac{\langle \delta^2 \rangle}{\left(\frac{p(m^c)}{p(m^*)}\right)^{\frac{2}{N}} N^{\frac{k^*-1}{N}} - 1}\right]^{1/2} \qquad (8)$$

where $\langle \delta^2 \rangle$ is now the variance of $m^*$ over the observation interval rather than in any particular dataset. For large $N \gtrsim 100$ and model description lengths of the true model such that $\Delta_M^* \equiv \mathcal{H}_M(m^*) - \mathcal{H}_M(m^c) \sim O(1)$, the transition noise is well approximated by ("Methods" section)

$$s_\epsilon^\times \approx \left[\frac{\langle \delta^2 \rangle N}{2\Delta_M^* + (k^* - 1)\log N}\right]^{1/2}. \qquad (9)$$

Therefore, since $s_\epsilon^\times$ diverges for $N \to \infty$, the true model is learnable for any finite observation noise, provided that $N$ is large enough (which is just another way of seeing that probabilistic model selection with the BIC approximation is consistent[11]).

In Fig. 3, we show that the upper bound in Eq. (8) is close to the exact transition point from the learnable to the unlearnable phase, as well as to the peak of the prediction error relative to the irreducible error. Moreover, once represented as a function of the scaled noise $s_\epsilon/s_\epsilon^\times$, the learnability curves of both models collapse into a single curve (Fig. 3e), suggesting that the behavior at the transition may be universal. Additionally, the transition between $m^*$ and $m^c$ becomes more abrupt with increasing $N$, as the fluctuations in $\langle \delta \rangle_D$ become smaller. If this upper bound became tighter with increasing $N$, this would be indicative of a true, discontinuous phase transition between the learnable and the unlearnable phases.

To further understand the transition and further test the validity of the assumption leading to the upper bound $s_\epsilon^\times$, we plot the average description length (over datasets $D$) of the MDL model at each level of observation noise $s_\epsilon$ (Fig. 4). We observe that the description length of the MDL model coincides with the description length $\mathcal{H}(m^*)$ of the true model below the transition observation noise, and with the description length $\mathcal{H}(m^c)$ of the trivial model above the transition observation noise. Around $s_\epsilon^\times$, and especially for smaller $N$, the observed MDL is lower than both $\mathcal{H}(m^*)$ and $\mathcal{H}(m^c)$, suggesting that, in the transition region, a multiplicity of models (or Rashomon set[18]) other than $m^*$ and $m^c$ become relevant.

## Discussion

The bidirectional cross fertilization between statistical learning theory and statistical physics goes back to the 1980's, when the paradigm in artificial intelligence shifted from rule-based approaches to statistical learning approaches[19]. Nonetheless, the application of statistical physics ideas and tools to learning problems has so far focused on learning parameter values[17,20–22], or on specific problems such as learning probabilistic graphical models and, more recently, network models[23–27]. Thus, despite the existence of rigorous probabilistic model selection approaches[28], the issue of learning the structure of models, and especially closed-form mathematical models, has received much less attention from the statistical physics point of view[10].

Our approach shows that, once the model-selection problem is formalized in such a way that models can be represented and enumerated, and their posteriors $p(m|D)$ can be estimated and sampled in the same way we do for discrete configurations in a physical system[5], there is no fundamental difference between learning models and learning parameters (except, perhaps, for the discreteness of model structures). Therefore, the same richness that has been described at length in parameter learning can be expected in model learning, including the learnability transition that we have described here, but perhaps others related, for example, to the precise characterization of the description length landscape in the hard phase in which generalization is difficult. We may also expect model-learning and parameter-learning transitions to interact. For example, there are limits to our ability to learn model parameters, even for noiseless data[17,29,30]; in this unlearnable phase of the parameter learning problem, it seems that the true model should also be unlearnable, even in this noiseless limit, something we have not considered here. Our findings are thus the first step in the characterization of the rich phenomenology arising from the interplay between data size, noise, and parameter and model identification from data.

Our results on closed-form mathematical models may also open the door to addressing some problems of learning that have so far remained difficult to tackle. In deep learning, for example, the question of how many training examples are necessary to learn a certain task with precision is still open[19], whereas, as we have shown, in our context of (closed-form) model selection the answer to this question arises naturally.

Finally, we have shown that, when data are generated with a (relatively simple) closed-form expression, probabilistic model selection generalizes quasi-optimally, especially in the ideal regime of low observation noise. This is in contrast, as we have seen, to what happens in this case with some machine learning approaches such as artificial neural networks. Conversely, it would be interesting to understand how expressive closed-form mathematical models are, that is, to what extent they are able to describe and generalize when data are not generated using a closed-form mathematical model (for example, for the solution of a differential equation that cannot be expressed in closed form). We hope that our work encourages more research on machine learning approaches geared towards learning such interpretable models[18], and on the statistical physics of such approaches.

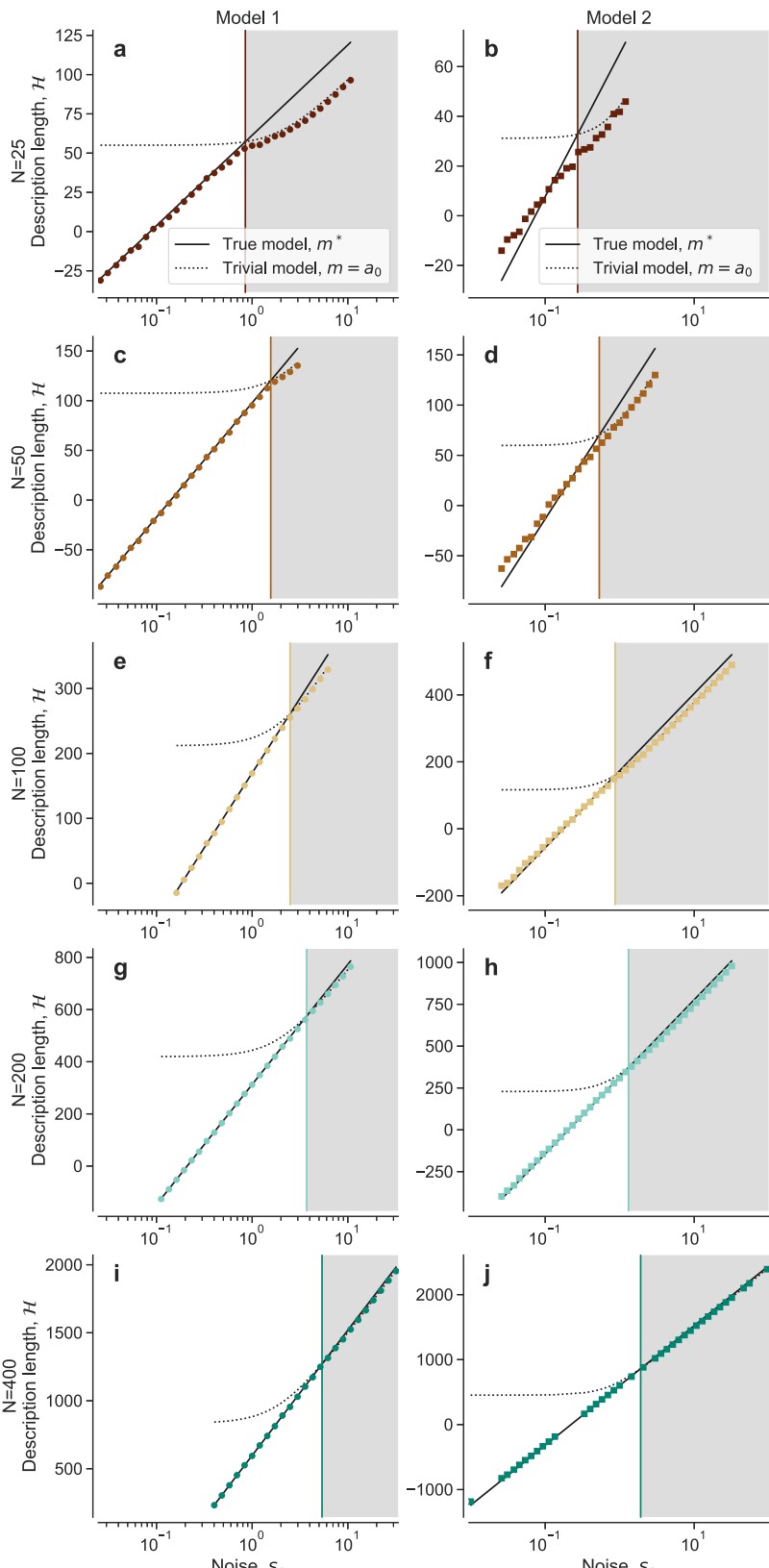

**Fig. 4 | Model description length and learnability conditions.** For each of the two models (**a**, **c**, **e**, **g**, **i** Model 1; **b**, **d**, **f**, **h**, **j**, Model 2) and each size $N$ (**a**, **b**, $N = 25$; **c**, **d**, $N = 50$; **e**, **f**, $N = 100$; **g**, **h**, $N = 200$; **i**, **j**, $N = 400$), we plot the description length of the MDL model identified by the Bayesian machine scientist, averaged over 40 realizations of the training dataset $D$ (colored symbols). For each model and $N$, we also plot the theoretical description length of the true generating model $m^*$ (Eq. (6); solid black line) and of the trivial model $m^c$ (Eq. (7); dotted black line). As in Fig. 3, colored vertical lines are estimates of the learnability transition point at which $\mathcal{H}(m^*) = \mathcal{H}(m^c)$ (Eq. (8)). Right of this point (gray region; unlearnable phase) $\mathcal{H}(m^*) > \mathcal{H}(m^c)$, so the true model cannot be learned, from the data alone, by any method.

## Methods

### Prior over models

The probabilistic formulation of the model selection problem outlined in Eqs. 1–4 requires the choice of a prior distribution $p(m)$ over models —although the general, qualitative results described in the body of the text do not depend on a particular choice of prior. Indeed, no matter how priors are chosen, some model $m^c = \arg\max_m p(m)$ (or, at most, a finite subset of models, since uniform priors cannot be used[5]) different from the true generating model $m^*$ will be most plausible a priori, so the key arguments in the main text hold.

However, the quantitative results must be obtained for one specification of $p(m)$. Here, as in the previous literature[5,8], we choose $p(m)$ to be the maximum entropy distribution that is compatible with an empirical corpus of mathematical equations[5]. In particular, we impose that the mean number $\langle n_o \rangle$ of occurrences of each operation $o \in \{+, *, \exp, \ldots\}$ per equation is the same as in the empirical corpus; and that the fluctuations of these numbers $\langle n_o^2 \rangle$ are also as in the corpus. Therefore, the prior is

$$p(m) \propto \exp\left[\sum_{o \in \mathcal{O}}(-\alpha_o n_o(m) - \beta_o n_o^2(m))\right], \qquad (10)$$

where $\mathcal{O} = \{+, *, \exp, \ldots\}$, and the hyperparameters $\alpha_o \geq 0$ and $\beta_o \geq 0$ are chosen so as to reproduce the statistical features of the empirical corpus[5]. For this particular choice of prior, the models $m^c = \arg\max_m p(m)$ are those that contain no operations at all, for example $m^c = \text{const.}$; hence the use of the term *trivial* to denote these models in the text.

### Sampling models with the Bayesian machine scientist

For a given dataset $D$, each model $m$ has a description length $\mathcal{H}(m)$ given by Eq. (4). The Bayesian machine scientist[5] generates a Markov chain of models $\{m_0, m_1, \ldots, m_T\}$ using the Metropolis algorithm as described next.

Each model is represented as an expression tree, and each new model $m_{t+1}$ in the Markov chain is proposed from the previous one $m_t$ by changing the corresponding expression tree: changing an operation or a variable in the expression tree (for example, proposing a new model $m_{t+1} = \theta_0 + x_1$ from $m_t = \theta_0 * x_1$); adding a new term to the tree (for example, $m_{t+1} = \theta_0 * x_1 + x_2$ from $m_t = \theta_0 * x_1$); or replacing one block of the tree (for example, $m_{t+1} = \theta_0 * \exp(x_2)$ from $m_t = \theta_0 * x_1$) (see ref. [5] for details). Once a new model $m_{t+1}$ is proposed from $m_t$, the new model is accepted using the Metropolis rule.

Note that the only input to the Bayesian machine is the observed data $D$. In particular, the observational noise $s_\epsilon$ is unknown and must be estimated, via maximum likelihood, to calculate the Bayesian information criterion $B(m)$ of each model $m$.

### Artificial neural network benchmarks

For the analysis of predictions on unobserved data, we use as benchmarks the following artificial neural network architectures and training procedures. The networks consist of: an input layer with two inputs corresponding to $x_1$ and $x_2$; (ii) four hidden fully connected feedforward layers, with 10 units each and ReLU activation functions; and (iii) a linear output layer with a single output $y$.

Each network was trained with a dataset $D$ containing $N$ points, just as in the probabilistic model selection experiments. Training errors and validation errors (computed on an independent set) were calculated, and the training process stopped when the validation error increased, on average, for 100 epochs; this typically entailed training for 1000–2000 epochs. This procedure was repeated three times, and the model with the overall lowest validation error was kept for making predictions on a final test set $D'$. The predictions on $D'$ are those reported in Fig. 1.

### Model description lengths

The description length of a model is given by Eq. (4), and the BIC is

$$B(m) = -2 \ln \mathcal{L}(m)|_{\hat{\theta}} + (k+1) \ln N, \qquad (11)$$

where $\mathcal{L}(m)|_{\hat{\theta}} = p(D|m, \hat{\theta})$ is the likelihood of the model calculated at the maximum likelihood estimator of the parameters, $\hat{\theta} = \arg\max_\theta p(D|m, \theta)$, and $k$ is the number of parameters in $m$. In a model-selection setting, one would typically assume that the deviations of the observed data are normally distributed independent variables, so that

$$\mathcal{L}(m) = \prod_i \frac{1}{\sqrt{2\pi s_y^2}} \exp\left[-\frac{(y_i - m_i)^2}{2s_y^2}\right]$$
$$= \frac{1}{(2\pi s_y^2)^{N/2}} \exp\left[-\sum_i \frac{(y_i - m_i)^2}{2s_y^2}\right] \qquad (12)$$

where $m_i \equiv m(\mathbf{x}_i)$, and $s_y$ is the observed error, which in general differs from $s_\epsilon$ when $m \neq m^*$. We can obtain the maximum likelihood estimator for $s_y^2$, which gives $s_y^2 = \frac{1}{N}\sum_i (y_i - m_i^*)^2$, and replacing into Eqs. (4)–(12) gives

$$\mathcal{H}(m) = \frac{N}{2} \ln 2\pi s_y^2 + \frac{N}{2} + \frac{k+1}{2} \ln N - \ln p(m). \qquad (13)$$

For $m = m^*$ we have that $s_y^2 = \frac{1}{N}\sum_i(y_i - m_i^*)^2 = \langle \epsilon^2 \rangle_D$. For any other model $m$, we define for each point the deviation from the original model $\delta_i := m_i - m_i^*$ so that $s_y^2 = \frac{1}{N}\sum_i(\epsilon_i - \delta_i)^2 = \langle \epsilon^2 \rangle_D + \langle \delta^2 \rangle_D - 2\langle \delta\epsilon \rangle_D$. Plugging these expressions into Eq. (13), we obtain the expressions in Eqs. (6) and (7).

### Approximation of $s_\epsilon^\times$ for large $N$

Defining $x = 1/N$ in Eq. (8) and using the Puiseux series expansion

$$a^{2x}\left(\frac{1}{x}\right)^{bx} = 1 + x\left(2\log a + b\log\frac{1}{x}\right) + O(x^2) \qquad (14)$$

around $x = 0$, we obtain Eq. (9).

## Data availability

We did not use any data beyond the synthetically generated data described in the manuscript.

## Code availability

The code for the Bayesian machine scientist is publicly available as a repository from the following URL: https://bitbucket.org/rguimera/machine-scientist/.

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

## Acknowledgements

This research was funded by project PID2019-106811GB-C31 from the Spanish MCIN/AEI/10.13039/501100011033 and by the Government of Catalonia (2017SGR-896).

## Author contributions

M.S.-P. and R.G. designed the research. O.F.-F., I.R., H.D.L.R., and R.G. wrote code and run computational experiments. O.F.-F., I.R., H.D.L.R., J.D., M.S.-P., and R.G. analyzed data. O.F.-F., I.R., M.S.-P., and R.G. wrote the manuscript.

## Competing interests

The authors declare no competing interests.
