## [Peer Review File · Nature Communications]

REVIEWER COMMENTS

Reviewer #2 (Remarks to the Author):

This manuscript addresses a fundamental question for the learnability of models from data: in the presence of finite datasets and noise, when can the generative process underlying the data be correctly inferred? Using a generic approach in a simplified setting, the authors report the existence of different regimes for the learnability of the model, including a size-dependent transition between learnable and unlearnable regimes. I believe these are significant results of broad interest because they connect directly to fundamental questions in Data Science and Machine learning. After the authors address the points below, in particular the question about the role of priors, I believe the manuscript can be published in Nature Communications.

1. Main concern. In footnote/reference [16] the authors state that "all results described below are independent of the choice of prior.". It is not clear to me in which extent this is true. In figure 2b, when some of the key results of the paper are described, the quantity being plotted is only depended on the prior. It is thus clear that there are underlying assumptions for the choice of the prior. One of them seems to be that simpler models have a higher prior probability. While this is reasonable, it is neither obvious nor necessary. It is thus essential to:

1a) Clarify what are the underlying choices of prior for which the results hold.

1b) Give a brief review about how priors are chosen in the "Machine Scientists", the readers will not all be familiar with this prior work and it is important for this manuscript to be self contained.

Specific points:

2. I believe that one strong underlying assumption is that the model used to generate the data is part of the set of models being considered in this analysis. This is typically not known or not an assumption (in Bayesian data analysis). I think it'd be important to indicate this point, possibly in the first paragraphs of the manuscript.

3. In Eq. (4), what are the assumptions for which the BIC approximation hold?

4. In Fig. 1, how are the x_1, x_2 values of the points chosen? What is the range of possible values?

Reviewer #3 (Remarks to the Author):

Dear authors,

Please find attached my report.

Best regards.

Reply to Reviewer #2

COMMENT: This manuscript addresses a fundamental question for the learnability of models from data: in the presence of finite datasets and noise, when can the generative process underlying the data be correctly inferred? Using a generic approach in a simplified setting, the authors report the existence of different regimes for the learnability of the model, including a size-dependent transition between learnable and unlearnable regimes. I believe these are significant results of broad interest because they connect directly to fundamental questions in Data Science and Machine learning. After the authors address the points below, in particular the question about the role of priors, I believe the manuscript can be published in Nature Communications.

We are very grateful for your careful reading of our manuscript, for your positive assessment of the work, and for your comments, which we address below and in the revised version of the manuscript.

COMMENT: Main concern. In footnote/reference [16] the authors state that “all results described below are independent of the choice of prior.” It is not clear to me in which extent this is true. In figure 2b, when some of the key results of the paper are described, the quantity being plotted is only depended on the prior. It is thus clear that there are underlying assumptions for the choice of the prior. One of them seems to be that simpler models have a higher prior probability. While this is reasonable, it is neither obvious nor necessary. It is thus essential to:

- 1. Clarify what are the underlying choices of prior for which the results hold.**
- 2. Give a brief review about how priors are chosen in the “Machine Scientists”, the readers will not all be familiar with this prior work and it is important for this manuscript to be self contained.**

We thank you for pointing out that this was not clear in the original submission. What we meant in the footnote was that none of the *qualitative* results in the paper depend on the choice of priors (optimality of the Bayesian model selection approach, existence of a transition, and possibility to derive analytical bounds for the transition point). As you note, some of the specific quantitative results that we show are indeed reliant on our particular choice of priors.

The key consideration here is that, no matter how priors $p(m)$ are chosen, some model $m^t = \arg \max_m p(m)$ (or, at most, a finite subset of models) different from

the true generating model m^* will be most plausible *a priori*.¹ As the noise in the data increases, and the data-dependent term $p(m|D)$ stops being able to clearly discriminate between m^* and all other models, the choice of model starts being driven by $p(m)$ —at this point, m^t becomes the most plausible/compressive model overall, and m^* becomes unlearnable.

In this manuscript, as well as in previous publications, we choose our prior as the maximum entropy distribution over models that is consistent² with an empirical corpus of mathematical equations (Guimera et al., Science Advances, 2019). As you mention, this leads to the most plausible models being the simplest ones (e.g. $m = \text{const}$ or $m = x_1$), which, as you also mention, is reasonable but not necessary. It also happens to be convenient in that it allows us to obtain a simple analytical expression for the description length of m^t and, thus, for the transition point where $p(m^*, D) = p(m^t, D)$. That being said, the fact that the crossover exists and the method to obtain the crossing point are independent of the specific choice of prior.

We have now clarified all of this in the manuscript. In particular, we have added a Methods section describing in detail our choice of priors, as well as a discussion on the qualitative (*vs* quantitative) generality of our findings. We have also added an analysis of a model not drawn from the prior and, indeed, quite implausible *a priori* (Supplementary text and Fig. S1; see also Fig. 1 below in our reply to Reviewer #3).

COMMENT: I believe that one strong underlying assumption is that the model used to generate the data is part of the set of models being considered in this analysis. This is typically not known or not an assumption (in Bayesian data analysis). I think it'd be important to indicate this point, possibly in the first paragraphs of the manuscript.

Indeed, we assume that the data have been generated using a closed-form mathematical model. This is crucial to the question we want to explore, namely, that *even when the data are generated from a closed-form model* it is not obvious that the true model can be identified from data. That being said, note that our approach includes *any* closed-form mathematical model—since the Bayesian machine scientist can explore (at least asymptotically) all closed-form mathematical expressions, the only assumption is that the true generating model can be written in closed form.

¹As discussed in the original publication on the Bayesian machine scientist, choosing a uniform prior is not an option because: (i) it is improper; (ii) there are infinitely many complex models than simple models and thus, for entropic reasons, choosing a uniform prior would lead to infinitely complex models for any dataset.

²In terms of the number of times that each operation occurs, and their variance.

In a practical situation in which we aimed to identify an expression for some arbitrary data set, things might be different—data could come, for example, from a differential equation whose solution cannot be expressed in closed form. Although we know that our approach can also find excellent approximations in those cases (Guimera et al, Science Advances, 2019; Reichardt et al., Physical Review Letters, 2020), studying the expressive power of closed-form mathematical models is beyond the scope of this manuscript. In any case, this is not very different from other contexts where Bayesian model selection is applied. Consider, for example, the use of Bayesian methods in network science for link prediction or to select the optimal partition of nodes into groups using stochastic block models (SBM). In that case, there is virtual certainty that the network has *not* been generated with the SBM; yet, SBM provides good approximations and, often, excellent results.

As per your suggestion, we have added a clarification along these lines in the introduction, as well as a longer sentence in the discussion about the ability of closed-form models to describe data generated by models that cannot be expressed in closed form.

COMMENT: In Eq. (4), what are the assumptions for which the BIC approximation holds?

The BIC results from using Laplace’s method to the integration of the distribution $p(D|m, \theta)p(\theta|m)$ over the parameters θ . Thus, the calculation assumes that: (i) the likelihood $p(D|m, \theta)$ is peaked around $\theta^* = \arg \max_{\theta} p(D|m, \theta)$, so that it can be approximated by a Gaussian around θ^* ; (ii) the prior $p(\theta|m)$ is smooth around θ^* so that it can be assumed to be approximately constant around θ^* . Whereas these assumptions are sometimes unjustified (for example, in network inference problems with the stochastic block model), in regression-like problems they are typically warranted.

We have clarified this in the manuscript.

COMMENT: In Fig. 1, how are the x_1, x_2 values of the points chosen? What is the range of possible values?

In Fig. 1 and throughout the manuscript, x_1 and x_2 are chosen uniformly at random in the interval $[-2, 2]$.

We have now clarified this in the caption of Figs. 1 and 2.

Reply to Reviewer #3

COMMENT: This work investigates the problem of inferring a statistical model from noisy data. Information theoretically, this problem is optimally solved by an exact estimation of the posterior distribution over the models given the data, which the authors approximate using a Monte Carlo sampling strategy called Bayesian machine scientist introduced in a work by a subset of the authors [1]. This estimation strategy is probed in a series of controlled experiments where data is generated from a true model (sometimes also referred to as planted or teacher-student setting). The main results in the manuscript are:

1. The minimum description length (MDL) model (known in the statistics literature as the maximum a posteriori (MAP) estimator) estimated from the “Bayesian machine scientist” achieves quasi-optimal generalisation error (error on fresh, unseen data).
2. The existence of a learnability transition as a function of the model noise and quantity of data available. Three regimes are identified: a learnable regime where the MDL/MAP coincides with the true model generating the data, an unlearnable regime where the MDL/MAP does not coincide with the true model and a transition regime where the MDL/MAP coincides with the true model for a fraction of the realisation of the problem.
3. An analytical upper bound for the critical noise below which the model is learnable, as a function of the number of samples, the size of the true model, description length gap and averaged reducible error of the trivial model.

We thank you for your careful reading of the manuscript and for your precise summary of the main results of our work. We also appreciate your thoughtful comments, which we hope are convincingly addressed below.

COMMENT: As a general comment, while the global message of the manuscript is clear, I found hard to grasp the core technical assumptions and details behind the results. For instance, it is not clear whether in the estimation of the posterior the authors use or not information about the model that generated the data. In the introduction the authors write:

When is it possible to identify m as the as the true generating model among all possible closed-form mathematical models, for someone who does not know the true model beforehand?

Which suggests the true model is not used during estimation. But in the discussion, the authors write:

In particular, the setup of our work here is the same as in other teacher-student learning scenarios [18], in which both prior and likelihood (of closed-form models, instead of parameters) are known.

Which suggests instead that the true model likelihood and prior are known to the statistician. Part of this confusion comes from the fact that the work heavily relies on an algorithm (“Bayesian machine scientist”) which is not discussed in the paper - not even on the Methods section. Although I understand that this algorithm is the subject of a published work, I would encourage the authors to consider adding a pseudo-code of the algorithm used together with a discussion of the main assumptions.

We now realize how these statements in the manuscript could seem contradictory and, thus, generate confusion; we apologize.

In our manuscript, we assume that the observations y_i of the dependent variable are generated as $y_i = m^*(x_i, \theta_{m^*}) + \epsilon_i$, where θ_{m^*} are the parameters of model m^* , and ϵ_i is a Gaussian, unbiased noise. We also assume that the model m^* can be written in closed-form, but we do not know anything else about it (for example, it could be $m^*(x, \theta) = \theta_0 + \theta_1 x_1$ or $m^*(x, \theta) = \sin(x_1) + \exp(\theta_1 x_2 - x_1) - \theta_2$ or anything else). **The functional form of the model m^* is what we aim to find** (not just the value of the parameters)—it is in this sense that we say that we “do not know the model beforehand.”

However, because the observational noise is Gaussian, we can write the likelihood of each functional form m as

$$p(y_i | m, \theta_m) = \frac{1}{\sqrt{2\pi s_\epsilon^2}} \exp\left(-\frac{(y_i - m(x_i, \theta_m))^2}{2s_\epsilon^2}\right). \quad (1)$$

It is in this sense (for each m , we can calculate its likelihood) that we wrote that we are “in a teacher-student scenario.”

This is not unlike what happens in “ordinary” learning settings in which one aims to learn the parameter values (for a fixed *model structure*), rather than the *model structure* itself, which is what we do here.³ Simply, the role played by parameters there is played by model structures m here.

We have clarified this in the manuscript and removed any potentially confusing reference to teacher-student scenarios. Following your suggestion, we have also

³Note that, in our setting, the parameters θ are eventually integrated out into the Bayesian information criterion, so our learning problem is about identifying m^* , not the true value of its parameters θ^* .

added a section to the Methods section describing in some detail the workings of the Bayesian machine scientist, which is an MCMC (Metropolis) algorithm that allows us to sample from the posterior $p(m|(y, x))$.

COMMENT: One of my main concerns is in how this work differs from the long line of works discussing the computational-to-statistical gaps (a.k.a. hard phase) in estimation problems [2, 3]. These works are based on an exact asymptotic analysis of the posterior distribution, so unless I am missing something I don't see how they can be different. However, the picture painted by [3] is richer, with the "learnability transition" (sometimes called "perfect recovery" in this line of work) being one among a plethora of possible behaviours in planted estimation problems, see e.g. [4] for a classification. If this is the case, the core of this work would be already contained in this literature.

Following up on our previous reply, we wish to emphasize that the setup of the problem is quite different from what is in the literature that you provide. There, the learning task is always about identifying the correct parameter values for a fixed (and known) model, whereas our work in this manuscript deals with identifying the correct (closed-form) model itself, m^* . Therefore, we believe that the core of this work is by no means contained in the literature.

That being said, we do expect that there are parallels with that literature. A case in point is the transition that we identify (defined by the crossing of model description lengths), which is akin to information theoretical transitions in "ordinary" learning problems (defined by the crossing of free energies). However, not all connections are so straightforward. In particular, many of the properties of the hard phase are a consequence of the structure of the configuration space of parameter values, which may or may not be shared by the (discrete and much more complex) space of closed-form model structures m . We expect that the study of these questions may lead to very important and general insights about model discovery/selection, but they are beyond the scope of this manuscript.

In any case, we have now expanded the framing and discussion of our results vis a vis the literature on learnability transitions and the hard phase in estimation problems.

COMMENT: In general, sampling for the posterior is computationally costly, and can become prohibitive when the model size is large.

How does the "Bayesian machine scientist" algorithm used in this work scale with the number of samples N and the model dimension? I noticed the data models used here are 2 dimensional. Would it be possible to run experiments of sizes comparable with the simplest modern learning tasks, say MNIST (784 dimensions) or CIFAR10 (3072 dimensions)?

The context in which the Bayesian machine scientist was developed is one in which one aims to identify interpretable, closed-form mathematical models m connecting a dependent variable $y = m(\mathbf{x})$ to a set of independent variables x_i , with $i = 1, \dots, k$. This, in fact, is the general setup in what is referred to, in different fields (and with slightly different meanings) as symbolic regression, computational equation discovery, or systems identification. Such approaches are applied, among others, to problems such as fluid mechanics (Reichard et al., Physical Review Letters, 2020), chemical kinetics (Pablo-García et al., ACS Catalysis, 2022), or even as tools to approximate unknown functions with the goal of performing analytical calculations (Artime & De Domenico, Nature Communications, 2021).

In these contexts, and taking into account that the focus is on interpretability and simple, closed-form mathematical models, one rarely deals with more than a handful of independent variables; virtually never more than $k = 10$.

In any case, the complexity of each MCMC step is determined by the calculation of the Bayesian information criterion, which scales as the number N of observations (x_i, y_i) (since it involves the calculation of mean squared errors). Increasing the dimension k of the problem has the effect of increasing the size of the space of possible models, which may slow down the convergence towards equilibrium models but has no effect on the computational complexity of the algorithm. In other words, for a fixed N , the number of models explored on a given time should not depend on the dimensionality k , although we may need more steps to sample the space of models properly.

COMMENT: How important is the additive i.i.d. Gaussian noise hypothesis here? Why can't one consider a general true model likelihood $P(y|m^*)$?

The assumption of i.i.d. Gaussian noise is standard in regression and symbolic regression problems, and it is what allows us to write the likelihood as in Eq. (1) above. In principle, we could assume other noise structures (for example, multiplicative noise) or, as you mention, even more general likelihoods, but these would be hard to justify in the context of regression and symbolic regression/model discovery. In any case, we do not expect that the qualitative behavior (optimality, transition, and so on) should change in any way from what we report in the manuscript.

We have clarified this in a new note in the introduction.

COMMENT: Is the discussion specific to the MAP estimator? How would it change if instead one considers other statistics of the posterior, say the posterior mean $m^{\text{MMSE}} = E[m|D]$?

In the context of model selection, and given that each m is a different closed-form mathematical model (again, one model could be $m_1(x, \theta) = \theta_0 + \theta_1 x_1$,

another could be $m_2(x, \theta) = \sin(x_1) + \exp(\theta_1 x_2 - x_1) - \theta_2$, we do not fully understand what you mean by $m^{\text{MMSE}} = E[m|D]$. Indeed, since the Bayesian approach is consistent, selecting the most plausible model amounts to using the MAP of $p(m|D)$.

When it comes to generalization to fresh, unseen data in the second section, things are different. There, the optimal prediction would come, indeed, from averaging over models. For example, we have that $y^{\text{MMSE}} = E[y|D]_m$, where the expectation is over the posterior $p(m|D)$ (see Guimera et al., Science Advances, 2020, for more details on this). However, in this section and throughout the present manuscript we focus on the problem of identifying the true generating model, so we limit ourselves to using the MAP throughout (see also our reply to your next comment below).

COMMENT: Probably that’s due to my confusion on what is assumed in the “Bayesian machine scientist” algorithm, but I don’t understand the point of Section II. Doesn’t the fact that the estimators generalise follow from the optimality of the posterior? Or the point of the section is to show that the approximation involved in the algorithm does not hurt generalisation?

Both, to some extent. Indeed, except for the (relatively mild) approximation in the calculation of the description length leading to the Bayesian information criterion, the posterior $p(m|D)$ results in optimal model selection. However, the optimal generalization to unseen data is achieved by averaging (integrating) over models, rather than by using the MAP of $p(m|D)$ alone. The purpose of the section is, thus, to show: (i) that the approximations in the calculation of the description length are warranted; (ii) that, even by using the most plausible model instead of averaging over models, our approach leads to quasi-optimal generalization (except in the region close to the transition).

Additionally, we fear that, especially within sectors of the computer science community (less so in physics), the optimality of the posterior and, more generally, of probabilistic approaches is not always fully appreciated. So, to be honest, we also wanted to dispel doubts and anticipate questions in this regard.

We have clarified this in the manuscript.

COMMENT: To elucidate better the connections with the literature (Q1) and the computational limitations of this work (Q2), an interesting point of comparison could be to see how the “Bayesian machine scientist” algorithm performs in a generalised linear estimation problem. As an example, take the real phase retrieval problem where the data is given by $m^*(x_i, \theta^*) = |x_i \cdot \theta^*|$ with $x_i, \theta^* \in R^d$. Bayes-optimal estimation for this problem has been extensively studied in [5, 6, 7], where it was shown that for i.i.d. Gaussian inputs

$x_i \sim \mathcal{N}(0, I_d/d)$ and weights $\theta^* \sim \mathcal{N}(0, I_d)$ the information theoretically θ^* can be reconstructed with $N \approx d$ samples, but computationally (with a first order algorithm) only with $N \approx 1.18d$. However, positive correlation (i.e. partial recovery) with θ^* can be achieved with $N \approx 0.5d$. Where does the learnability transition lie here? How does the “Bayesian machine scientist” perform for $d = 1000$ and $N = 250, 600, 1100, 1500$?

The generalized linear estimation problem that you propose is quite different from the symbolic regression problem that we tackle in the manuscript (by means of the Bayesian machine scientist). Indeed, what we aim to identify is the **functional form** of m^* , rather than the values of the parameters θ^* . Additionally, our observed data are noisy, so for any finite data size N , we observe the learnability transition at a finite variance of the observation noise s_ϵ . In other words, the size N alone does not determine the transition, but rather the balance between N and s_ϵ .

In any case, we have now used your request to hopefully clarify our contribution. Following your suggestion, we have generated data using the generalized linear model $m^*(x, \theta^*) = |x \cdot \theta^*|$, so that individual, noisy observations are given by $y_i = |x_i \cdot \theta^*| + \epsilon_i$, with $\epsilon_i \sim \mathcal{N}(0, s_\epsilon)$. For the reasons discussed above, we do not use $d = 1000$ but rather $d = 5$, which would be considered more suitable within the context of symbolic regression. Other than this, data sets are generated exactly as suggested, with $x_i \sim \mathcal{N}(0, I_d/d)$, weights $\theta^* \sim \mathcal{N}(0, I_d)$, and $N \in \{50, 600\}$. Additionally, we generate different data sets for each level of noise s_ϵ .

Again, the learnability question we are interested in is whether it is possible to identify the true generating model $m^* = |x \cdot \theta|$ (regardless of the precise value of the parameters), as opposed to competing models such as $m = (x \cdot \theta)^2$, $m = |x_1\theta_1 + x_2\theta_2 + \theta_3|$, $m = x \cdot \theta$, or any other expression. As we show in Fig. 1 below (and new Fig. S1 in the manuscript), the learnability transition for this model is exactly as for the 2 models reported in the manuscript.⁴ Note also that we are well beyond the information theoretic transition for the learnability of *model parameters* since $N \gg d$, and yet we still observe the learnability transition at finite values of the observation noise. This transition thus happens in a region that is far from the information theoretical transition where the values of the parameters themselves become unlearnable.

We have now added this analysis to the manuscript as supplementary information. We have also clarified that, even in the noiseless case, the ability of our approach to detect the correct model is limited by the inability to learn the parameter values in the transition described in your references [5, 6, 7].

⁴Note that this is true even though this model is not drawn from the prior $p(m)$. Indeed, this model is quite implausible a priori, according to our maximum entropy priors.

Figure 1: Learnability transition for a generalized linear estimation problem, where the true generating model is $y_i = |\theta^* \cdot x_i| + \epsilon_i$, with $\theta^* \in \mathbb{R}^d$ and $x_i \in \mathbb{R}^d$, $d = 5$. We generated data using $\epsilon_i \sim \mathcal{N}(0, s_\epsilon)$, $x_i \sim \mathcal{N}(0, I_d/d)$, weights $\theta^* \sim \mathcal{N}(0, I_d)$, and $N \in \{50, 600\}$. Even though this model is not directly drawn from the prior and, in fact, has low a priori probability, and even though points are generated differently from the examples the body of the paper, the transition occurs exactly as in those examples and as predicted by our theory.

COMMENT: **The authors repeat at several points in the text (pages 1, 2-3, 6) that:**

This is in contrast to standard machine learning approaches, which generate very complex models that, as we show, are suboptimal in the region of low observation noise.

This assertion is too generic and misleading. How standard ML methods (here taken broadly as empirical risk minimisation on parametric models) compare to Bayes-optimal estimation is subtle, and depends on the data distribution and model class. This question has been investigated for generalised linear models for different data models, e.g. in a teacher-student framework [8, 9] and for Gaussian mixture data [10, 11], where it has been shown that in some cases optimally regularising your model (say by cross-validation) is enough to reach Bayes-optimal performance. In these cases, ERM is a much more efficient way to reach optimal performance than posterior sampling.

We agree that this statement was too generic. We have modified the text where necessary to make it clear that the limitations we are discussing are restricted to the particular problem we study here.

REVIEWER COMMENTS

Reviewer #2 (Remarks to the Author):

The addressed have addressed all my comments, I recommend the manuscript for publication.

Reviewer #3 (Remarks to the Author):

Dear authors,

First let me thank you for carefully answering my questions and for welcoming my suggestions. I think most of my confusion has been cleared by your reply. However, there are still some points I would like to clarify.

1. As you probably have noticed, the difference between parameter estimation vs. model/symbolic estimation did not come across very clearly in my first reading. While the first paragraph of the section "I. Discussion" stresses this difference, I think it could be stressed earlier in the manuscript, for instance in the introduction. This would help the reader who is more familiar with the former rather than the latter.

Of course, despite the conceptual distinction, the whole philosophy of modern machine learning methods such as neural networks is not very far from "model estimation". Indeed, choosing a huge parametric model with large model complexity means one can potentially learn very different functional models (i.e. feature learning and model adaptivity). Exploring the space of closed-form mathematical expressions weighted by a plausability measure (i.e. choice of prior) with the "Bayesian machine scientist" is not very different from finding a basis (i.e. features) in the space of functions with universal approximators such as deep neural networks.

For this reason, I find the comparison proposed in the section "Artificial neural network benchmarks" quite interesting. However, choosing a very narrow network for the targets in Fig. 1 might not be a fair comparison (can you reasonably approximate these models with 10 relu units?). The "phase retrieval" target I proposed in my review might be a fairer comparison, since the absolute value function can be actually learned with 2 units.

2. I appreciate the authors have added more details on the "Bayesian machine scientist" algorithm. But overall I am still confused about it. In particular, beside the data D , what are the inputs to the algorithm? Crucially, does it know the ground truth noise level s_{ϵ} or not? Please clarify this also in this new section.

3. Related to my previous "Question 2". How does the BIC criterion exactly scale with the number of observations N (e.g. linearly, quadratically, exponentially, etc.)? And how the number of possible models scale with the dimension k ? Finally, how the MC mixing time scales with the number of possible models?

I fully understand that your motivation is small k where these questions might not be very relevant. However, I think these are crucial questions for a paper which proposes to solve a problem from computer science. If small k is a bottleneck of the proposed methodology (which it seems to be the case), it is better to be fully transparent about it.

Reply to Reviewer #2

COMMENT: **The authors have addressed all my comments, I recommend the manuscript for publication.**

We thank you for your constructive comments and for your recommendation.

Reply to Reviewer #3

COMMENT: **First let me thank you for carefully answering my questions and for welcoming my suggestions. I think most of my confusion has been cleared by your reply. However, there are still some points I would like to clarify.**

We thank you again for your constructive feedback, both in the first round and now. We are happy that our answers and revision of the manuscript have cleared most of your questions.

COMMENT: **As you probably have noticed, the difference between parameter estimation vs. model/symbolic estimation did not come across very clearly in my first reading. While the first paragraph of the section “I. Discussion” stresses this difference, I think it could be stressed earlier in the manuscript, for instance in the introduction. This would help the reader who is more familiar with the former rather than the latter.**

Following your suggestion, we have now emphasized this in the introduction as well as at the beginning of the results section, when we formulate the problem probabilistically.

COMMENT: **Of course, despite the conceptual distinction, the whole philosophy of modern machine learning methods such as neural networks is not very far from “model estimation”. Indeed, choosing a huge parametric model with large model complexity means one can potentially learn very different functional models (i.e. feature learning and model adaptivity). Exploring the space of closed-form mathematical expressions weighted by a plausability measure (i.e. choice of prior) with the “Bayesian machine scientist” is not very different from finding a basis (i.e. features) in the space of functions with universal approximators such as deep neural networks.**

For this reason, I find the comparison proposed in the section “Artificial neural network benchmarks” quite interesting. However, choosing a very narrow network for the targets in Fig. 1 might not be a fair comparison (can you

reasonably approximate these models with 10 relu units?). The “phase retrieval” target I proposed in my review might be a fairer comparison, since the absolute value function can be actually learned with 2 units.

This is, we agree, an interesting test. Following your suggestion, we have now studied the performance of the benchmark artificial neural networks (ANNs) on the phase retrieval problem that you proposed. The results (now added to Fig. S1 in the Supplementary Materials and shown below) confirm that the accuracy of the ANN on fresh data is limited in the low-noise regime.

Figure 1: Learnability and extrapolation ability of probabilistic model selection and ANNs.

This cannot be a problem of insufficient expressiveness of the ANN—as you point out, a few hidden units are enough to model the true data perfectly, in principle. Rather, we argue that this is caused by *too much* expressiveness. In the low-noise regime, the BMS is almost certain to identify the correct model, and thus it interpolates optimally between observations in the training set—only the correct model is considered. By contrast, the ANN has a lot of flexibility to interpolate, that is, it finds many acceptable ways to interpolate between the observed points. Thus, in this region, the accuracy of the ANN is not limited by the noise in

the data but by the density of points in the training set—lower density means more possibilities to interpolate.

COMMENT: I appreciate the authors have added more details on the “Bayesian machine scientist” algorithm. But overall I am still confused about it. In particular, beside the data D , what are the inputs to the algorithm? Crucially, does it know the ground truth noise level s_ϵ or not? Please clarify this also in this new section.

Data D is the only input to the algorithm. The ground truth noise level s_ϵ is unknown to the Bayesian machine scientist; it is estimated from D using the maximum likelihood estimator, as prescribed by the calculation of the Bayesian information criterion. We have clarified this in the new section as requested.

COMMENT: Related to my previous “Question 2”. How does the BIC criterion exactly scales with the number of observations N (e.g. linearly, quadratically, exponentially, etc.)? And how the number of possible models scale with the dimension k ? Finally, how the MC mixing time scales with the number of possible models?

I fully understand that your motivation is small k where these questions might not be very relevant. However, I think these are crucial questions for a paper which proposes to solve a problem from computer science. If small k is a bottleneck of the proposed methodology (which it seems to be the case), it is better to be fully transparent about it.

The BIC is extensive in the number of observations N . Its calculation involves estimating the values of the parameters of the model by least squares, which in general is a difficult problem whose solution is not guaranteed for most models m . In practice, the MCMC generates new models m_{i+1} by changing the currently sampled model m_i (with parameters θ_i). Therefore, estimating the parameters θ_{i+1} of a newly generated model, m_{i+1} , from those of the preceding model, θ_i , is typically much easier than in the worst case. In any case, each iteration of the least squares procedure is of order N .

With regards to number of possible mathematical expressions for a given k , the dependency is (in the worst case) k^L , where L is the number of leaves of the tree that represents the expression. Thus, the number of possible mathematical expressions grows as a (large) power of the number of features. Finally, we do not have a precise estimation of the mixing time, but we observe large variability for different models, even for fixed k . In particular, the Markov chain will typically thermalize quickly for simple models (for example, a linear model) even for a relatively large number of features, whereas it will take longer for a complex model with few features.

We have now clarified upfront in the introduction that we focus on the the regime in which the dimension of feature space is low compared to some other machine learning settings.

REVIEWERS' COMMENTS

Reviewer #3 (Remarks to the Author):

I thank the authors for welcoming the suggestions and for their rebuttal. I can confirm they have addressed all my questions and concerns in this revised version. Therefore, I can recommend the manuscript for publication.

Reply to Reviewer #3

COMMENT: I thank the authors for welcoming the suggestions and for their rebuttal. I can confirm they have addressed all my questions and concerns in this revised version. Therefore, I can recommend the manuscript for publication.

We thank the reviewer again for their constructive feedback during the whole reviewing process.